# Simplified Optimization of the Magnetic Configuration of HEMP-Thrusters

Lars Lewerentz * and Ralf Schneider *

Institute of Physics, University of Greifswald, Felix-Hausdorff-Str. 6, D-17489 Greifswald, Germany
* Correspondence: lars.lewerentz@uni-greifswald.de (L.L.); schneider@uni-greifswald.de (R.S.)

**Abstract:** One type of ion thruster that has gained attention in recent years is the High Efficiency Multistage Plasma (HEMP) thruster. Optimizing the performance of these thrusters can be challenging due to the complexity of the underlying physics. Since the construction of new designs is expensive, cheaper methods for optimization, e.g., numerical optimization, are being sought. This paper presents a fast, analytical approach to finding realistic starting points for the magnetic geometry design of HEMP thrusters. First, a ratio of length to radius is presented, where the magnetic field is especially parallel at the center of the magnetic ring. This result is confirmed with the open-source library magpylib. Its speed and accuracy qualify this tool for further optimization processes. Here, we present some simple performance indicators, which may be beneficial to characterize the magnetic field structure for further optimization.

**Keywords:** ion thruster; optimization; magnetic field

## 1. Introduction

Ion thrusters are electric propulsion systems that use plasma to generate thrust. These thrusters have several advantages over traditional chemical propulsion systems, making them an attractive option for use in space. One of the main advantages of ion thrusters is their high specific impulse, which is a measure of the efficiency of a propulsion system. Their specific impulses can be an order of magnitude higher than chemical propulsion systems, allowing them to achieve much greater velocities and travel much farther with the same amount of propellant mass. This makes them ideal for long-duration missions, such as those of interplanetary spacecraft, where the weight and volume of the propellant is a major concern. Because of such high specific impulses, they require much less propellant to generate the same amount of thrust as a chemical propulsion system. This means that they can operate for much longer periods of time without needing to refuel.

One type of ion thruster that has gained attention in recent years is the High-Efficiency Multistage Plasma (HEMP) thruster, developed and patented by Thales Deutschland GmbH [1]. It uses a specific configuration of magnetic fields using "cusps" to control and accelerate the plasma. A "cusp" is a region where the magnetic field lines converge. In HEMP thrusters, the cusp magnetic fields are created by ring-shaped permanent magnets of opposed polarity. Usually, a thruster consists of two, three, or more of these ring magnets.

A schematic view of a HEMP-Thruster is presented in Figure 1. HEMP thrusters consist of a cylindrical discharge chamber, surrounded by several permanent magnet rings with pairwise opposed poles. This induces a cusp-like magnetic field structure in the channel, acting as a magnetic mirror. The inside of the chamber is coated with a dielectric, which has a high sputtering threshold to minimize erosion, for example. boron nitride. The anode and the feed gas inlet are located at the bottom of the channel. Usually, xenon is used as feed gas, because of its high mass and its properties as a noble gas. An electron-emitting cathode, the neutralizer, is located outside the channel exit, attached to the thruster. It feeds the discharge in the channel and neutralizes the plume. Neutralizing the positively

charged exit stream of xenon ions keeps the satellite from charging up with a positive charge. The positive potential at the anode leads to a plasma building up in the thruster channel. With sufficient electric conductivity of the plasma, the potential drop is moved from the anode to the thruster exit, where the plasma density decreases. This creates an electric field in front of the thruster exit that accelerates the electrons toward the thruster. There, the electrons follow the magnetic field lines provided by the permanent magnets. The resulting gyro radius of the electrons is much smaller than the channel radius, resulting in the magnetization of the electrons. Due to the magnetic field structure, they remain trapped in the thruster channel. The confinement of the electrons leads to high ionization rates. Because of the magnetization of the electrons and the magnetic field line structure between the cusps, the electron transport parallel to the symmetry axis is strong inside the channel. In contrast, the ions are not magnetized due to their large mass and the magnetic field strength typical for such thrusters (≈200 mT), and drift with low energies through the thruster, following the distribution of the electrons. At the exit, they are accelerated by the electric field resulting from the potential drop. They are emitted with high emission velocities, thus generating thrust. This mode of operation, with high ionization efficiency, high exit velocity of the ions, and minimal erosion, makes the HEMP-Thruster a very attractive electric propulsion device. It can generate thrust in a wide range from 1 μN to 100 mN. The flexibility to provide thrust over a wide range and its long lifetime due to minimal erosion make the HEMP-Thruster a favorable concept for long-duration space missions [2]. More information on HEMP thrusters can be found in [3]. One advantage of the HEMP-Thruster is that it is relatively simple to operate, as it does not require the use of high-voltage grids or other complex systems. This makes it more reliable and easier to maintain than other types of ion thrusters. However, the HEMP-Thruster requires a lot of power to operate, and the cost of the permanent magnets can be high.

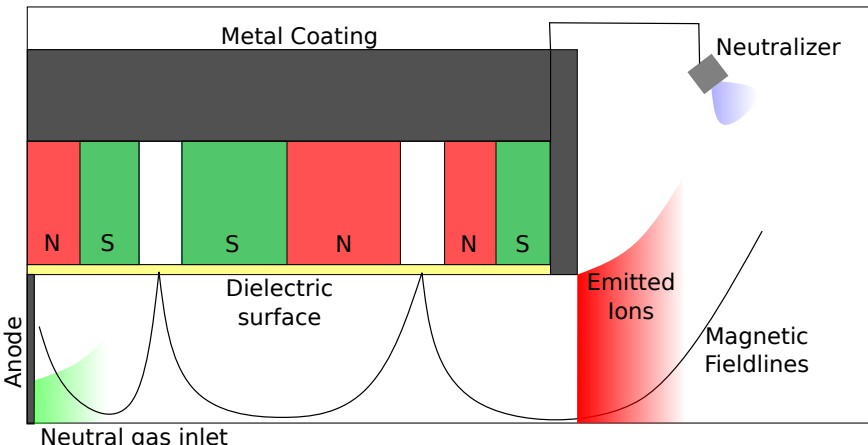

**Figure 1.** Schematic view of a three-stage HEMP-Thruster. The cylinder is cut horizontally along the cylinder's axis of symmetry.

Optimizing the performance of these thrusters can be challenging due to the complexity of the underlying physics and the need for precise control of the ionized gases. A key aspect of optimization is the design of the ionization chamber, where the neutral propellant is ionized into plasma. This chamber should be designed to maximize the ionization rate while minimizing the amount of energy required to ionize the propellant. Another important aspect is the design of the acceleration stage, where the ions are accelerated to generate thrust. This stage should be designed to maximize the acceleration of the plasma particles while minimizing the amount of energy required to accelerate the ions. Additionally, the amount of neutral particles that are not ionized and thus accelerated should be minimized, as these particles can impede the performance of the thruster. Finally, optimizing the overall performance of HEMP thrusters requires careful control of the operating conditions, including the temperature and density of the plasma, the electric fields used to accelerate

the plasma, and the magnetic fields used to confine the plasma. These operating conditions should be carefully controlled and monitored to ensure that the thruster is operating within the optimal range for maximum performance.

The optimization of ion thrusters is a difficult process and revolves around experimental measurements [4]. Optimizing the performance of HEMP thrusters requires a multi-disciplinary approach that encompasses the design of the ionization chamber, the acceleration stage, and the overall operating conditions. By carefully designing and controlling these aspects of the thruster, it is possible to achieve significant improvements in performance, including higher specific impulse and lower propellant consumption. Since the construction of new designs is expensive, cheaper methods for optimization are sought-after. In [5–7] a HEMP-Thruster geometry was numerically optimized by a genetic algorithm. Even using simplified power balance models with kinetic corrections requires a lot of computational effort for such an optimization, because the parameter space of such a search is huge. It is beneficial to characterize some traits by other means in order to speed up the costly numerical optimization and to have the ability to improve other properties.

This paper presents a fast, analytical approach to finding realistic starting points for the magnetic geometry of HEMP-Thruster design optimization, which can then be used for further refined optimization. This approach to calculating magnetic fields is not only much faster, but also has reduced error compared to numerical approaches, which is particularly important for transport codes where artificial drifts and forces are minimized.

First, there is an analytical estimate for the best geometry of a magnetic stage, where the magnetic field should be parallel to the symmetry axis at the center. It will be shown that this estimate is very close to the experimental optimization of two-stage thrusters by trial and error [8,9]. A similar ratio was also found in the rather complex and time-consuming optimization procedure by genetic algorithms [7].

Secondly, using the analytical solutions further diagnostics can be motivated by physics. Electrons are magnetized and follow the field lines. Therefore, long field lines combined with the confinement by the magnetic mirror effect at the cusps will increase the ionization and thereby the efficiency of the thruster. The different geometries are compared with respect to the field lengths by field line tracing. For the analysis, a weight function reflecting the observed density distribution is used. To obtain a global quality measure for the different magnetic field configurations, a geometric scaling of the single field lines is applied. The mirror effect at the end of the cylindrical magnets determines the electron confinement by limiting the losses and is responsible for enhanced ionization.

## 2. Materials and Methods

The magnetic structure is analyzed analytically at the center of a magnetic stage. A parallel field between the cusps is a desirable feature since it allows for high mobility of electrons in the axial direction. Hence, axial potential gradients are small. Since electrons are kept away from the wall except at the cusps, radial fields exist only there. Then, electrons are able to ionize the plasma efficiently, so that ignition is facilitated. Electrons, which enter the channel at the symmetry axis $r = 0$, can overcome the cusp so that the following stages are also populated.

A simplified analytical expression on the axis is derived. The analytical result is used to validate the Python package magpylib [10]. This uses a generalized analytical method for the calculation of magnetic fields. The field line lengths of the magnetic flux **B** are then calculated numerically. Next, their length is labeled $\ell$ and measured in mm. The Python library matplotlib [11] is used for plotting, and the scipy library is employed for field line integration [12].

### 2.1. Analytical Estimate

Throughout the paper, the symmetry of the cylindrical device is used. A HEMP-Thruster consists of several stages of ring magnets with repelling orientation to each other. In this paper, one stage will be studied, but the extension to more stages can be performed.

The natural coordinate system consists of the radial $r$, azimuthal $\varphi$, and axial $z$ coordinates. Figure 2 shows the $(r,z)$-plane of the magnetic geometry for a typical stage of a HEMP-Thruster. The $z$-axis is the symmetry axis. A rotation around the $z$-axis yields the three-dimensional stage. Terms proportional to $\partial_\varphi$ and the magnetic field component $B_\varphi$ are zero due to the symmetry.

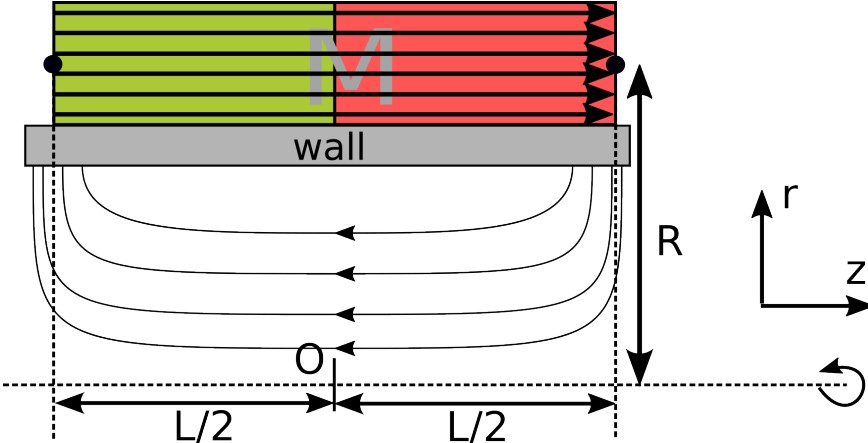

**Figure 2.** $(r,z)$-plane of the magnetic geometry for a typical stage of a HEMP-Thruster. The center is denoted by the origin (O). The radius $R$ and the half-length $L/2$ are also shown.

Vector quantities are denoted in bold letters. The starting point for the optimization is to find the ratio of length $L$ to radius $R$ where the magnetic flux **B** is homogeneous between both ends of the magnet. In other words, we look for an optimal geometry of $L$ and $R$, where the field at the origin is as parallel as possible along the $z$-axis. The end regions form, together with other stages, the cusp regions where the wall contact happens. Hence, the condition of a homogeneous field in the middle of one stage supports the design goal to limit the wall contact as much as possible to the cusps.

For the analytical analysis, the magnetization **M** of the magnet is assumed constant. In the magnetostatic case, Maxwell's equations reduce to

$$\nabla \times \mathbf{H} = 0 \Leftrightarrow \mathbf{H} = -\nabla \Phi_m$$
$$\nabla \mathbf{B} = 0 \, ,$$

where $\Phi_m$ is a magnetic potential, **B** is the magnetic induction and $\mu_0$ is the vacuum permeability. **B** is linked with the magnetic field **H** and **M** by

$$\mathbf{B} = \mu_0 (\mathbf{H} + \mathbf{M}) \, .$$

The two equations for the magnetic field produce a Poisson equation for the magnetic potential

$$\nabla^2 \Phi_m = \nabla \mathbf{M} \, ,$$

where the sources are the divergence of the magnetization. This equation can be treated in the same way as the electrostatic Poisson equation [13]. Instead of charges, the divergence of the magnetization is the source on the right-hand side. In this application, the sources of the potential are the front faces of the magnets, where the constant magnetization changes at the boundary to the surrounding media. For an infinitesimally thin cylinder, the calculation of the potential on the axis of the stage ($r = 0$) can be simplified to

$$\Phi_m(z) = \frac{\mu_0 M R}{4\pi} \left( \frac{1}{\sqrt{R^2 + (z + L/2)^2}} - \frac{1}{\sqrt{R^2 + (z - L/2)^2}} \right), \tag{1}$$

where $R$ and $L$ are the radius and length of the magnetic stage, respectively. Here, the front faces are reduced to a point source. A Taylor's series expansion of Equation (1) at the origin ($z = 0$) up to the fifth order results in the numerator

$$
\begin{aligned}
\Phi_m(z) \propto 16\,&\left(L^5 - 20\,L^3R^2 + 30\,LR^4\right)z^5 \\
+\,4\,&\left(L^7 + 2\,L^5R^2 - 32\,L^3R^4 - 96\,LR^6\right)z^3 \\
+\,&\left(L^9 + 16\,L^7R^2 + 96\,L^5R^4 + 256\,L^3R^6 + 256\,LR^8\right)z\,.
\end{aligned}
$$

The calculation is performed with the open-source mathematics software system SageMath [14]. A corresponding notebook can be found on GitHub [15], where the data is made public. In order to have a magnetic field **H**, which is parallel to the symmetry axis up to the highest order, the fifth- and first-order terms of the potential shall remain. The $z^3$ term must vanish, i.e., the polynomial in front needs to be zero, so that the resulting field is parallel up to the fourth order. The roots of the polynomial are

$$
L = -2i\,R,\ L = 2i\,R,\ L = -\sqrt{6}R,\ L = \sqrt{6}R \text{ and } L = 0\,.
$$

Since $R$ and $L$ are the geometric dimensions of the magnet, we are only interested in the real and positive solutions, so that only the geometric ratio

$$
L = \sqrt{6}R \tag{2}
$$

is an adequate solution. The potential contains linear, cubic, and fifth-order terms in $z$. By finding the root in front of the cubic term, the solution is left with only linear and fifth-order terms. A multipole expansion off-axis in $r$ with the help of Legendre polynomials specifies the $r$ dependency in the vicinity of the origin. The linear term is then canceled and only a fifth-order term in Legendre polynomials remains. The calculation is found for the electrostatic case in [13]. Due to the expansion, the potential has terms depending on $r$ and $z$. The removal of the linear and cubic terms thus minimizes the field in $r$ as well.

This ratio for a single stage of length to radius $L/R \approx 2.45$ produces a homogeneous magnetic field on the axis. It is remarkable that a rather complex optimization procedure [5–7], finite element methods, power balance equations, and even kinetic corrections based on Particle-in-Cell (PIC) calculations reached a similar value. Furthermore, the experimentally optimized industry designs operate close to this parameter choice [8,9].

### 2.2. Numerical Analysis of Field Lines

For a set of radius $R$ and length $L$ of the permanent magnetic ring, the magnetic field is evaluated at $400 \times 400$ points in the $(r,z)$-plane. The magnetic field of a ring magnet of finite width of 2 mm is calculated by the Python library magpylib [10] and is used to verify the results. The Jupyter notebooks are hosted on [15]. The library can solve the magnetostatic problem of a cylinder [16,17] with magnetization **M** analytically. Two cylinders of different polarity and radius but the same strength result in a ring-shaped magnet, where the smaller cylinder is inscribed within the larger one. This is allowed due to the superposition of magnetic "charges" $\nabla\mathbf{M}$. The problem is equivalent to an electrostatic case with a charge density at the front surface. The strength of the magnetization $\mu_0 M$ is set to 1 T, which is about the order of rare earth magnets.

This approach is very attractive for optimization because finite element solvers for magnetic fields require much longer run times. In addition, the conservation properties of the resulting fields (divergence-free, etc.) are better with analytical methods. This minimizes numerical errors from such fields, e.g., in PIC codes [18,19]. Divergent contributions from numerical field solvers are a big concern for such codes, because they produce artificial drifts and forces.

In Figure 3a–c the field lines of the magnetic flux **B** for three different geometries (R,L) are plotted. The origin of the coordinate system is the same as in the sketch of the analytical derivation. The three geometries have varying lengths (a) $L = 10$ mm, (b) $L = 50$ mm, and

(c) $L = 90$ mm with fixed radius $R = 20$ mm. The magnetic flux **B** is depicted as a stream plot in blue. For the different lengths of the ring magnet, the extent of the field lines in axial direction varies greatly. A marker (★) tags the maximum $z$ coordinate of the separatrix $z_{sep}$ in Figure 3, which will be discussed in more detail. In order to better understand the distribution of field lines, each plot contains 20 selected **B** field lines, which are traced from the starting points $(r_0, z = 0)$ towards the wall. These field lines are displayed in red. The initial radial position $r_0$ is varied in order to scan the whole domain, which is later used for the measures of the magnetic field **B**. Here, only half of the ring magnet is shown, since the field is also symmetric with respect to the $z = 0$ axis.

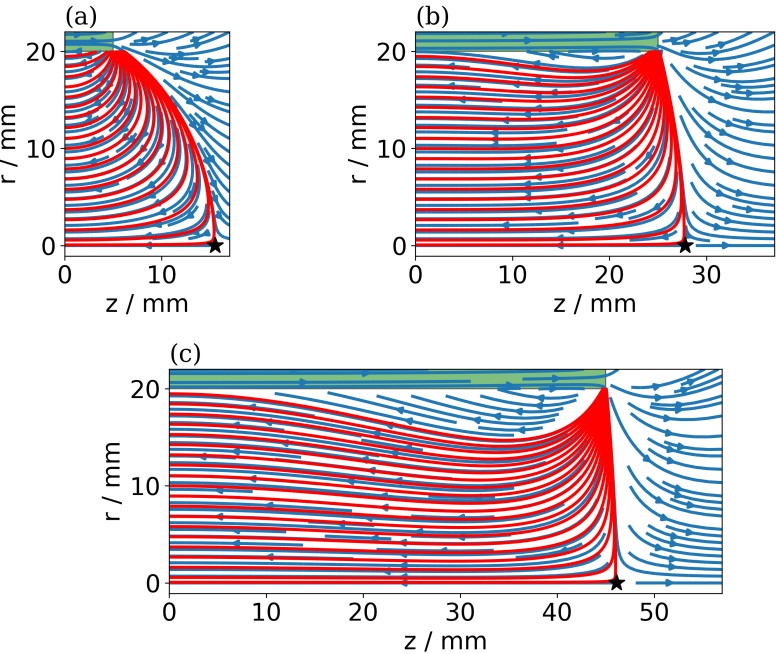

**Figure 3.** A ring magnet (green rectangle) is depicted with radius $R = 20$ mm for different lengths $L$ centered around the symmetry axis ($r = 0$). The three geometries have varying lengths (**a**) $L = 10$ mm, (**b**) $L = 50$ mm and (**c**) $L = 90$ mm. In each plot, the field lines are drawn in blue. The figure shows only one half of the axially symmetric system, neglecting negative $z$. Red lines start from various radii $r_0$, where $z = 0$. These are integrated along the magnetic field until they reach the wall, i.e., when they exceed R. The star (★) denotes the maximum $z$ coordinate of the separatrix $z_{sep}$ on the axis $r = 0$.

The tracing of the field lines was formulated as a system of discretized ordinary differential equations (ODE) where the vector of the magnetic field **B** is the vector of the derivative so that the ODE solver of the Python scipy package can be used. The paths of the 20 field lines, which are picked, are integrated until they hit an ideal wall at $r = R$. For that purpose, the field between the analytical grid points was interpolated linearly. Field lines that exceed the length $L/2$ of the thruster ring also stop at the ideal wall $r = R$. Normally, the magnetic rings are covered by dielectrics and metals so that the radial and axial extent is usually larger. This justifies the ideal wall condition.

For a starting point close to the axis, a ratio of radial to the total magnetic field $\sin(\alpha) = B_r/B$ is averaged over analytical sampling points in the vicinity of the origin within the box area ($r \leq 0.1 \cdot R$, $z \leq 0.1 \cdot L$). The result is shown in Figure 4. Close to the center of the ring magnet, the radial magnetic field $B_r$ is small compared to the total magnetic field B. The zero radial field, equivalent to only axial parallel magnetic field, is reached in agreement with the analytical result derived before; i.e., the root of $\sin(\alpha)$ is very close to the analytical result, denoted by the olive vertical bar in the figure. The minor deviation of the root from the analytical result is attributed to the finite width of the cylinder in the numerical calculation.

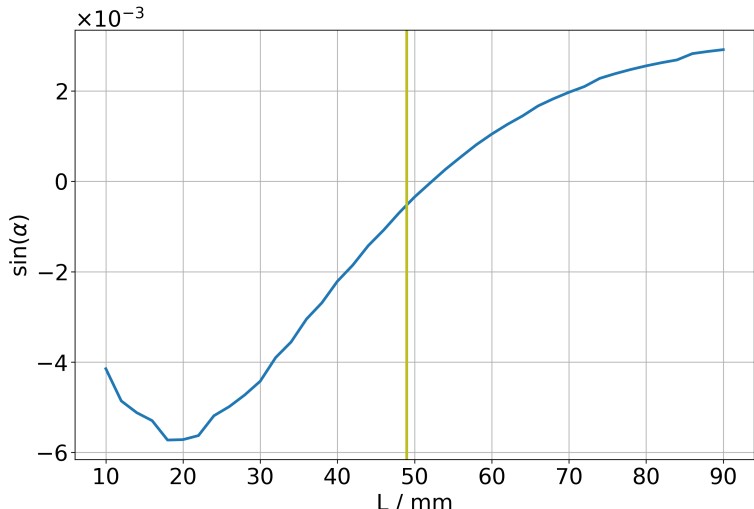

**Figure 4.** Ratio of the radial to the total magnetic field $\sin(\alpha) = B_r/B$ averaged over analytical sampling points within the box area ($r \leq 0.1 \cdot R$, $z \leq 0.1 \cdot L$,) for various configurations $L$ and fixed radius $R = 20$ mm. The olive vertical line denotes the analytical solution, where the field lines are particularly parallel.

## 3. Results

After the validation of the package, further numerical diagnostics can be used to characterize the properties of the magnetic configurations with respect to different aspects of physics. In the following, different geometries are calculated, where the length $L$ varies between 10 mm and 90 mm in 2 mm increments for a fixed radius $R = 20$ mm. For each geometry, the field and 20 selected field lines are computed. Visually, this procedure is transcribed in Figure 3. Two measures are introduced in order to characterize the geometry. The subsequent figures demonstrate different properties of the magnetic flux **B**.

Electrons in the HEMP-Thruster are magnetized and follow the field lines. Therefore, long field lines and the confinement by the magnetic mirror effect at the cusps increase the ionization and thereby the efficiency of the thruster. The different geometries are compared with respect to the field lengths by field line tracing.

The geometries shown in Figure 3 illustrate the field configuration change when the length $L$ is varied. One observes different properties of the magnetic field for the short geometry, Figure 3a, compared to intermediate, Figure 3b, or long magnets, Figure 3c. Relative to the length of the ring magnet, the field lines protrude more into the vacuum outside the magnet for smaller magnets. A measure $\tilde{z}$ of how far the concerning field lines elongate along the $z$ axis is plotted in Figure 5. The $z$ coordinate at the axial symmetry ($r = 0$) where the axial magnetic field $B_z$ changes direction defines the position $z_{sep}$ of the separatrix of the magnetic field. Thus, a relative distance $\tilde{z}$, how far the separatrix moves from the channel axially, is defined with respect to the channel length $(z_{sep} - L/2)/L$. The position of $z_{sep}$ has been marked for the three cases in Figure 3 with a black star ($\star$). For the three panels, the position relative to the exit $\tilde{z}$ varies greatly with length $L$. The blue curve $\tilde{z}(L)$ in Figure 5 displays the relation for the whole scan. For smaller lengths, the relative elongation increases quickly. Longer designs have a nearly negligible protrusion. The exact placement influences and shapes the exit field of a HEMP-Thruster, which in turn will influence the angular distribution of the expelled ions and consequently the thrust [20].

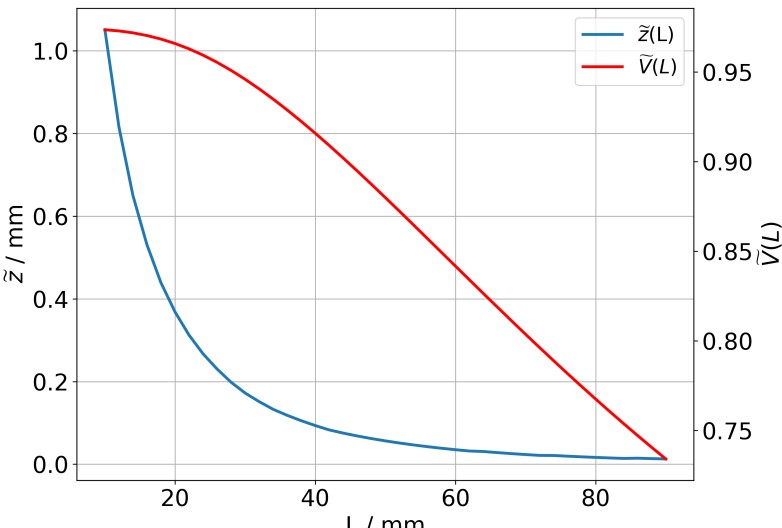

**Figure 5.** The relative distance of how far the separatrix elongates axially $\widetilde{z} = (z_{sep} - L/2)/L$ and the ratio $\widetilde{V}$ of the active volume $V_{active}$ to the total volume $V_{total}$ as a function of $L$ are plotted. Both affect the performance of a HEMP thruster.

A second observation from the field line plots in Figure 3 is the appearance of regions that are not accessible from field lines starting at the center of the cusp. These regions will have less filling by parallel transport and will depend strongly on radial transport effects. This is normally detrimental, but having such a small region here can even be beneficial for a HEMP-Thruster, because it can further reduce the wall contact of the plasma just to the cusp region, which is one of the design goals. The region that is not accessed by the field lines is called the passive region, whereas the region covered by field lines is called active. The active volume $V_{active}$ is calculated as the volume spanned by the closest diagnostic field line to the axis. The ratio $\widetilde{V}$ of the active volume to the total volume within the magnetic ring is also displayed in Figure 5. It is a measure of the volume that will be filled by electrons due to parallel transport along field lines connected to the near-axis region.

The passive volume is filled by short field lines without contact with the central near-axis region. Here, the radial transport of electrons from the central region populates this region slowly, and therefore, the electron density is reduced. The active volume decreases with the increasing length of the magnet and the passive volume increases, as expected from the magnetic field plots, which can be seen in Figure 3.

The integration of several field lines, which correspond to the red lines in Figure 3a–c, at equidistant radii $r_0$ delivers a length scale measure $\ell$ of the magnetic field **B**, which is shown in the upper panel of Figure 7. The field line length is nearly proportional to the length $L$. For small magnet lengths, the lengths $\ell$ deviate from a linear dependence, since the bigger elongation in the $z$ direction adds to the path. The average of the 20 field lines seems to be a good indicator for the qualitative behavior of $\ell$. In future studies, the average alone might be sufficient for an optimization procedure.

Further analysis is performed by a weight function reflecting the observed density distribution. A typical HEMP electron density distribution $n_e$ of a PIC simulation for a xenon discharge is plotted in Figure 6. The code for the simulation was developed within the group. Further details about the physics and the PIC code can be found in [21]. PIC calculations can deliver a better understanding of the processes in HEMP.

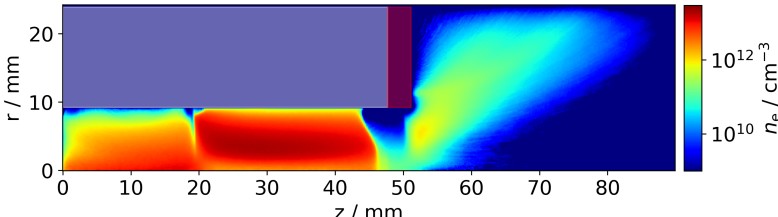

**Figure 6.** Electron density $n_e$ from a 2D PIC simulation of a two-stage DM3a HEMP-Thruster in the $(r,z)$-plane. The rectangular regions in gray and brown indicate regions with dielectric and metal wall boundaries, respectively. Particles are not able to penetrate into these areas.

The cusps show up as density sinks in the electron density in Figure 6 because the plasma–wall interaction is concentrated at these locations. They are clearly visible at the ends of the channel and at the cusp at z about 20 mm. The cusps act as magnetic mirrors and trap the electrons in between, such that the ionization probability is strongly enhanced, and larger densities in between the cusps are thereby obtained. For an axial position $z_0$ between the cusps, a radial decay in density is observed due to the individual length of the field lines. The higher the radial position $r_0$, the lower the electron density becomes. The weight function used in the following analysis is motivated by the density distribution obtained by PIC, which decays linearly and radially outwards in the logarithmic plot of Figure 6, so that for the contribution of the particle density, an exponential factor $\exp -r_0$ is assumed. Because of the cylinder symmetry, field line numbers starting at the radial coordinate $r_0$ scale with the circumference and are $r_0$ times more abundant. The mirror effect at the end of the cylindrical magnets determines the electron confinement by limiting the losses and is responsible for enhanced ionization. It is calculated as the ratio of the magnetic fields at the starting point $B_0 = B(z_0, r_0)$ to the magnetic field at the endpoint $(z_w, r_w)$ of the field line close to the wall $B_{max} = B(z_w, r_w)$

$$R_m = B_{max}/B_0 \,.$$

The ratio of the velocity components yields $\sin(\theta) = 1/\sqrt{R_m}$ of the velocity perpendicular to the magnetic field $v_\perp$ to the total velocity $v$. Then, the ratio $\sin(\theta)$ gives the reflection condition $\sin(\theta) \geq 1/\sqrt{R_m}$ [22,23]. Magnetic field lines with a high magnetic ratio are likely to reflect electrons moving along these toward the cusps. The time $\tau$ of the electrons spent between the mirrors is $\tau = l/v \propto l \cdot \sqrt{R_m}$. A confinement time weighted by volume and geometry results in

$$\tau = \ell \cdot \sqrt{R_m} \cdot r_0 \cdot e^{-r_0} \,. \tag{3}$$

Figure 7 shows individual field line lengths $\ell$ for three selected field lines and the average of 20 lines. The magnetic field lines' lengths nearly scale linearly with the length of the magnet. Deviations occur for small lengths. Here, field lines protrude further outside the channel (cf. Figures 3 and 5). From the twenty field lines, three are chosen in order to understand the different behaviors of a field line for a certain geometry. In addition, the average of the selected 20 field lines is also displayed. Trivially, magnetic field lines close to the symmetry axis $r = 0$ are longer than field lines with a higher radius $r_0$. The difference in length is also nearly linear with respect to $r_0$. This becomes more complicated for the two other quantities. The magnetic mirror ratio grows non-linearly, since the distance between the starting and end points of the field lines grows. In the bottom panel, the confinement time $\tau$ is presented. The time increases with longer channel lengths. Due to the radially decaying plasma density, the confinement times vary significantly for different starting positions.

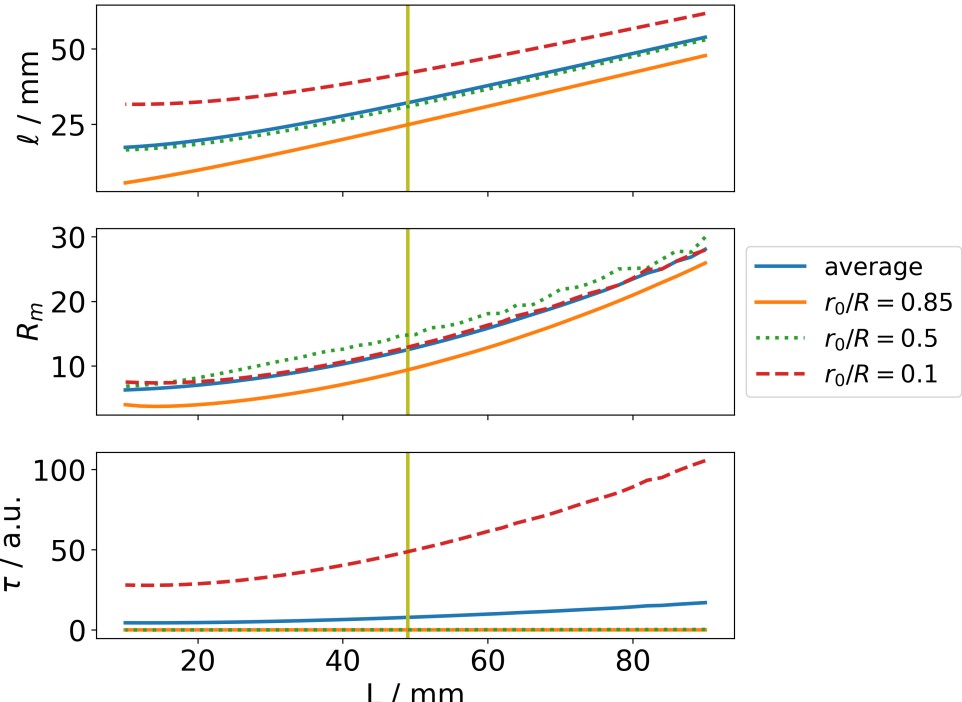

**Figure 7.** Field line properties are scanned over a range of magnet lengths $L$ and for a fixed radius $R = 20\,\text{mm}$. From top to bottom, the field line length $\ell$, the magnetic mirror $R_m$, and a confinement time $\tau$ from Equation (3) are plotted. Shown are the average lengths over 20 field lines (in blue) and three selected field lines for comparison. The time $\tau$ is in arbitrary units (a.u.).

In Figure 8, the same quantities as in Figure 7 are plotted. This time, $L$ is fixed at 50 mm and $R$ is varied between 6 mm and 40 mm, also in steps of 2 mm. Again, the field line length grows with increasing dimensions of the magnetic ring. The magnetic mirror $R_m$ does not scale linearly with $R$. A minimum of the magnetic ratio is visible. The exact location of the minimum for the scan varies for different field lines $r_0$. For instance, the field line average has a minimum at about 20 mm. The radial distance of the starting positions to the magnetic pole becomes important with increasing radius $R$ so that the mirror ratio decreases. With an even larger radius, the axial spread at the wall of the radially fixed starting positions increases geometrically. This leads to a larger area of the field lines at the magnetic pole and thereby a magnetic field **B** with a strongly changing magnitude. For the average magnetic ratio, the losses due to the diverging magnetic field at the cylinder end first lead to a decrease in the magnetic mirror. For greater radii, this is compensated by the distance of the starting position of the magnetic field.

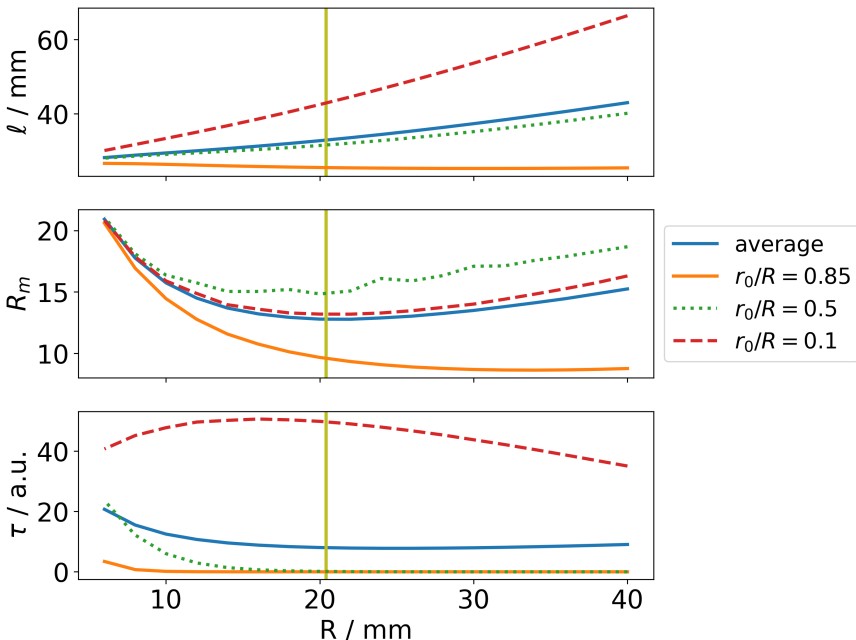

**Figure 8.** Same as Figure 7, but here, radius $R$ is varied instead of the fixed length $L = 50\,\text{mm}$. Since $R$ changes, the three selected field lines are plotted for fixed ratios of $r_0/R$.

## 4. Conclusions

Optimization of the magnetic field geometry for a HEMP-thruster is very important for improved thruster operation. The use of an analytical Python package allows for simplified and fast optimization. The package was validated by comparison with the specific solution at the center of the cusp, where a particularly parallel magnetic field was derived at the origin. In experiments, two-stage thrusters were optimized by trial and error to reach a length-to-radius ratio of about 5 to 1. This means that the length-to-radius ratio for one stage is about 2.5 to 1, which is very close to the analytical optimization result of 2.45. The geometric relation can also be seen in Figure 6 and in [8,9] for comparison. A similar ratio was also found in the optimization procedure using genetic algorithms [7].

Further diagnosis is then numerically possible, motivated by physics. The length of the magnetic field lines was calculated numerically in a scan of the ring geometry, which is linked to the path lengths of the electrons. In addition, density weights from experiments were used for the field line analysis, and the mirror effect at the end of the cylindrical magnets was diagnosed. This is important for electron confinement. Confining the electrons between the cusps and minimizing radial losses to the walls improves ionization.

The combination of different aspects of the optimization of the physics can be implemented very quickly and flexibly by defining a penalty function with different weightings for the different contributions. This allows numerical optimization of the magnetic field (including manufacturing constraints for the magnets) within minutes instead of days or weeks. Further combinations with power balance models or kinetic corrections can be made from this initial configuration, minimizing the risk of becoming stuck in local minima during the optimization process.

An optimizing strategy for the magnetic field **B** is as follows:

- Start from the optimal $R$ and $L$ of the analytical derivation.
- Vary $R$ and $L$. $R$ is expected to get a little smaller; see Figure 8. The magnetic mirror ratio becomes higher, and thus the confinement time.
- $L$ is expected to increase, which is the same statement as the previous point (cf. Figure 5). The intention is to avoid pushing the separatrix outward, which in turn may increase the exit angle of the ions. This is especially important for the exit stage.

- The increase in $L$ is limited by the reduction in the active volume $\widetilde{V}$ in Figure 5, which significantly reduces the electron density in these regions.

  In general, this simplified approach reduces the dimension of the remaining optimization problem. The calculation of magnetic fields is not only much faster but also has reduced error compared to common numerical approaches, which is particularly important for transport codes where artificial drifts and forces are minimized.

**Author Contributions:** Conceptualization, L.L. and R.S.; methodology, L.L. and R.S.; software, L.L.; formal analysis, L.L. and R.S.; writing—original draft preparation, L.L.; writing—review and editing, R.S.; visualization, L.L. All authors have read and agreed to the published version of the manuscript.

**Funding:** This work is funded by the European Union as EU project HEMPT-NG2 [grant id 101004140].

**Data Availability Statement:** The code for the numerical analysis can be found here: https://github.com/laleph/thruster_optimization (accessed on 20 February 2023).

**Conflicts of Interest:** The authors declare no conflict of interest. The funders had no role in the design of the study; in the collection, analyses, or interpretation of data; in the writing of the manuscript; or in the decision to publish the results.

## Abbreviations

The following abbreviations are used in this manuscript:

| | |
|---|---|
| MDPI | Multidisciplinary Digital Publishing Institute |
| DOAJ | Directory of open access journals |
| HEMP(-T) | High Efficient Multistage Plasma (Thruster) |
| PIC | Particle-in-Cell |
| ODE | Ordinary Differential Equations |

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
