# Peer review of "Simplified Optimization of the Magnetic Configuration of HEMP-Thrusters"

_applsci, doi:10.3390/app13063491_

Round 1

Reviewer 1 Report

In this work, the authors propose a very interesting semi-analytical approach to optimize the configuration of high efficiency multistage plasma thrusters. Their method seems to work, and using it to obtain initial parameters might greatly reduce the computational time required for running multiple PIC simulations to perform parameter scans. However, the manuscript is rough in many aspects. There might be errors (typos, probably) in Eq. 3 (parameters a and d are mentioned in the text that are not present in this equation) and/or Eq. 4 (I did not find the same expression when expanding Eq. 3 in the same Taylor series the authors mentioned). The figures are not tightly connected to the text, or are not referred to in the same order they are displayed. There is no visual consistency (while some figures use very small fonts, other have huge tick labels), and some legend keys are overlapping almost half of the panels in which they were inserted. Hence, in my opinion, although authors proposed an interesting motivation, in order to have this manuscript published, following comments must be clarified and responded in a satisfying way.

[Figure 1] Figure 1 is not mentioned in the text. Every figure should be addressed in the text.

[lines 105-108] "The magnetic structure is analyzed analytically at the center of a magnetic stage. A parallel field at the axis is good for the confinement of electrons minimizing radial wall 106 contact and allows easy ignition, because electrons from the neutralizer can enter the 107 thruster channel at the symmetry axis r = 0.".

[line 131] After writing the Maxwell's equations that are relevant for your problem, you are describing their variable and parameters, as usual. However, you also includes M and \mu_0 in this description, as if they were used in the aforementioned equations. Please fix the text (for example "Being M the magnetization, and \mu_0 the vacuum permeability, B is linked with H..."), or introduce M and \mu_0 after Eq. 1 (in which they are used). In addition, while in the equations you adopt bold letters to write vector quantities, in the text you are writing them without using bold letters. Please keep consistency when writing vector quantities, always using bold letters (unless you are specifically addressing their modulus).

[line 133] Rather than writing "Vector quantities are denoted in bold letters" here, please move this statement right after its first use (or even before it).

[line 142] After Eq. 3, you wrote "where a and d are the dimensions of the magnetic stage ... The constant c is proportional to the divergence of M.". However, Eq. 3 does not have such quantities. What is wrong: the text, or the equation?

[line 146] Using Mathematica to expand Eq. 3 in a Taylor series at z=0, I do not get the same expression you wrote in Eq. 4. Although I found a similar term for z^5, the other terms (for z^3 and z) are quite different. Please show me the detailed calculation you used to obtain Eq. 4.

[line 146] "In order to have a magnetic field H, which is parallel to the symmetry axis up to fourth order, the z^3 must vanish". Please explain why this condition is enough to ensure that H will be parallel to the symmetry axis. In addition, after showing a series expansion to the 5th order in Eq. 4, why did you chose to ensure the symmetry to the fourth order?

[line 149] Again, you are using a and d without having used them in any equation.

[line 157] The previous figure mentioned in the text was Figure 2 (line 120). Now, without any mention to Figures 3, 4, and 5, you jumped to Figure 6. Please call all figures in the text, and order them accordingly.

[line 173] "In Figure 3 (a-c) the field lines for three different geometries (R,L) are plotted.". Since you are dealing with multiple distinct vector fields, please always be specific regarding which field you are referring to. This will greatly improve the readability of your work. Please observe that, along the document, there are multiple instances in which you write "the field lines", rather than "the X field lines", where X is the field you are referring to (for example, B or H).

[lines 173-177] Although you added three panels in Fig. 3, you have not added any comparison or discussion regarding their characteristics. Again, each figure (and each panel of it) should be discussed in the text.

[Figure 3] While each panel individually looks good, with clear lines and colors, their arrangement as a three-panel figure is very messy. The panels have a large white border (panel c white border is overlapping the legends of panels a and b); the font size of panels a and b is much smaller than that of panel c (all panels in the same figure should have the same font size); while the labels of panels a and b are close to their bottom, there is a huge white space between panel c bottom and its label (probably because of the very wide white border around it). In the figure caption, ensure to be specific when referring to a field ("The field lines are blue": which field? "Red lines start from...": which field are they representing?).

[lines 193-195] "After the validation of the package, further numerical diagnostics can be used to characterize the properties of the magnetic configurations with respect to different physics aspects.". Before reading these lines, I was not aware that the package was validated. In the previous section, although you explained your methodology, and showed results in Figure 3, you did not mention anything about the validation. Please consider adding a paragraph on this matter.

Moreover, still on the matter of validating your approach, since you have access to PIC codes and knowledge on how to use them, why you did not usem to validate your semi-analytical approach? For example, you could compare the magnetic field lines obtained by using your approach (for example, those shown in Fig. 3) with magnetic field lines from PIC simulations performed for the same configurations.

[lines 200-201] "Comparing the different magnetic fields, one observes a protruding of the magnetic field into the vacuum outside the magnet.". Is this a comparison between Fig. 3 panels a, b, and c? If yes, please write this explicitly.

[lines 201-202] "The z coordinate at the axial symmetry (r = 0) where the axial magnetic field Bz changes direction is a measure for the position zsep of the separatrix of the magnetic field.". I strongly suggest you showing z_sep in Fig. 3 panels (a, b, and c), for example, by adding on them a large point labeled z_sep.

[Figures 4 and 5] The font size in these figures is huge compared to the others. I suggest you adopting the same font sizes and styles for all figures.

[lines 218-220] "The length L is varied between 10mm and 90mm in 2mm increments for a fixed R=20mm. Similarly, R is varied between 6mm and 40mm, also in steps of 2mm, while L is fixed at 50mm.For each of these cases, the field and the integration of the 20 field lines are performed, resulting in the subsequent figures.". Again, please address explicitly which figure depicts which case.

[lines 234-241] Is Figure 6 originated from the same simulation that originated Figure 5 of Ref. 21? I understand that all details can be obtained from it ("Further details can be found in [21]"). However, I suggest you at least mentioning in this manuscript the PIC code you used for the simulation.

[Figures 7 and 8] Besides having a huge font size, the legend keys overlapping half panel in these figures is a very poor choice. Please fix this.

[Conclusions] Please check the English/grammar writing in this section (some sentences and words are not correct). 

Author Response

Dear Reviewer,

We are very grateful for your suggestions and helpful comments. The recommendations really helped us improve the text, its readability and logical structure.

Answers to your questions can be found here. Most of the text changes are highlighted in blue. Please see the attached PDF for further details.

In this work, the authors propose a very interesting semi-analytical approach to optimize the configuration of high efficiency multistage plasma thrusters. Their method seems to work, and using it to obtain initial parameters might greatly reduce the computational time required for running multiple PIC simulations to perform parameter scans. However, the manuscript is rough in many aspects. There might be errors (typos, probably) in Eq. 3 (parameters a and d are mentioned in the text that are not present in this equation) and/or Eq. 4 (I did not find the same expression when expanding Eq. 3 in the same Taylor series the authors mentioned). The figures are not tightly connected to the text, or are not referred to in the same order they are displayed. There is no visual consistency (while some figures use very small fonts, other have huge tick labels), and some legend keys are overlapping almost half of the panels in which they were inserted. Hence, in my opinion, although authors proposed an interesting motivation, in order to have this manuscript published, following comments must be clarified and responded in a satisfying way.

Answer:

The notation of equation 3 was changed at short notice in order to be more consistent with the second part of the paper. "a" and "d" have been renamed to R and L/2, respectively.
For deriving eq. 4 we used the open-source mathematical system SageMath. A notebook with the derivation can be found on the GitHub data repository.
Be aware, we changed the numbering of the equations. Only equations, which are referred to directly in the text are numbered.

[Figure 1] Figure 1 is not mentioned in the text. Every figure should be addressed in the text.

Answer:

Now, the figure is referred to in the text. The other figures have been mentioned accordingly.
"A schematic view of the HEMP thruster is presented in Fig. 1."

[lines 105-108] "The magnetic structure is analyzed analytically at the center of a magnetic stage. A parallel field at the axis is good for the confinement of electrons minimizing radial wall 106 contact and allows easy ignition, because electrons from the neutralizer can enter the 107 thruster channel at the symmetry axis r = 0.".

Answer:

The text is extended and clarified. (lines 111-117)
"A parallel field between the cusps allows for a high mobility of electrons in axial direction. Hence, axial potential gradients are small.  Since electrons are kept away from the wall except at the cusps, radial fields exist only there.  Then, electrons are able to ionize the plasma efficiently, so that ignition is facilitated. Electrons, which enter the channel at the symmetry axis r = 0 can overcome the cusp so that the following stages are also populated.116For this, the simplified analytical expression at the origin is derived. "

[line 131] After writing the Maxwell's equations that are relevant for your problem, you are describing their variable and parameters, as usual. However, you also includes M and \mu_0 in this description, as if they were used in the aforementioned equations. Please fix the text (for example "Being M the magnetization, and \mu_0 the vacuum permeability, B is linked with H..."), or introduce M and \mu_0 after Eq. 1 (in which they are used). In addition, while in the equations you adopt bold letters to write vector quantities, in the text you are writing them without using bold letters. Please keep consistency when writing vector quantities, always using bold letters (unless you are specifically addressing their modulus).

Answer:

The vector quantities are consistently denoted in bold letters and the notation is explained.

[line 133] Rather than writing "Vector quantities are denoted in bold letters" here, please move this statement right after its first use (or even before it).

Answer:

The statement has been moved towards the beginning of the section.

[line 142] After Eq. 3, you wrote "where a and d are the dimensions of the magnetic stage ... The constant c is proportional to the divergence of M.". However, Eq. 3 does not have such quantities. What is wrong: the text, or the equation?

Answer:

The text is wrong. It has been adapted accordingly.

[line 146] Using Mathematica to expand Eq. 3 in a Taylor series at z=0, I do not get the same expression you wrote in Eq. 4. Although I found a similar term for z^5, the other terms (for z^3 and z) are quite different. Please show me the detailed calculation you used to obtain Eq. 4.

Answer:

We used the freely available tool SageMath to calculate the Taylor expansion. A Jupyter notebook is uploaded to the GitHub data page. We have added the reference.

[line 146] "In order to have a magnetic field H, which is parallel to the symmetry axis up to fourth order, the z^3 must vanish". Please explain why this condition is enough to ensure that H will be parallel to the symmetry axis. In addition, after showing a series expansion to the 5th order in Eq. 4, why did you chose to ensure the symmetry to the fourth order?

Answer:

The text has been clarified. (lines 152-155)
"In order to have a magnetic field H, which is parallel to the symmetry axis up to highest order, the fifth and first order term of the potential shall remain. The z^3 term must vanish, i.e., the polynomial in front needs to be zero, so that the resulting field is parallel up to fourth order."

[line 149] Again, you are using a and d without having used them in any equation.

Answer:

The notation is fixed. The text is updated.

[line 157] The previous figure mentioned in the text was Figure 2 (line 120). Now, without any mention to Figures 3, 4, and 5, you jumped to Figure 6. Please call all figures in the text, and order them accordingly.

Answer:

We missed to insert a figure, which is now Fig. 4. The old figures 4 and 5 have been combined to a new Fig. 5 in order to be more concise. The old figures contained only one simple measure.

[line 173] "In Figure 3 (a-c) the field lines for three different geometries (R,L) are plotted.". Since you are dealing with multiple distinct vector fields, please always be specific regarding which field you are referring to. This will greatly improve the readability of your work. Please observe that, along the document, there are multiple instances in which you write "the field lines", rather than "the X field lines", where X is the field you are referring to (for example, B or H).

Answer:

Thank you very much for the suggestion. We will be more precise. The field lines correspond to the magnetic flux B. The text is adapted, where needed.

[lines 173-177] Although you added three panels in Fig. 3, you have not added any comparison or discussion regarding their characteristics. Again, each figure (and each panel of it) should be discussed in the text.

Answer:

Indeed we added a necessary description. Thank you for pointing out. Here, we tried to be brief. Later, the geometric measures are explained in more detail. The exemplary field line plots show the limiting cases of the field line protrusion and active volume. When we are discussing the measures we reference them to the field line plots in Fig. 3.

[Figure 3] While each panel individually looks good, with clear lines and colors, their arrangement as a three-panel figure is very messy. The panels have a large white border (panel c white border is overlapping the legends of panels a and b); the font size of panels a and b is much smaller than that of panel c (all panels in the same figure should have the same font size); while the labels of panels a and b are close to their bottom, there is a huge white space between panel c bottom and its label (probably because of the very wide white border around it). In the figure caption, ensure to be specific when referring to a field ("The field lines are blue": which field? "Red lines start from...": which field are they representing?).

Answer:

The old figure consists of three subfigures. Apparently, the Latex settings put the single figures on top of each other. Now, we generate the figure directly and include it as a single piece. As a bonus, the labels have the same font size.

[lines 193-195] "After the validation of the package, further numerical diagnostics can be used to characterize the properties of the magnetic configurations with respect to different physics aspects.". Before reading these lines, I was not aware that the package was validated. In the previous section, although you explained your methodology, and showed results in Figure 3, you did not mention anything about the validation. Please consider adding a paragraph on this matter.

Answer:

The missing Figure 4 (in the new counting) compares the ratio of the radial field to the total magnetic field B. The root of the ratio function sin(alpha) averaged in the box area over L is very similar to the analytical optimum. This supports the analytical estimate and the numerical scan, since they both predict a similar minimum for the varied length L of the radial magnetic field B_r.

Moreover, still on the matter of validating your approach, since you have access to PIC codes and knowledge on how to use them, why you did not use to validate your semi-analytical approach? For example, you could compare the magnetic field lines obtained by using your approach (for example, those shown in Fig. 3) with magnetic field lines from PIC simulations performed for the same configurations.

Asnwer:

In our PIC codes we use a magnetic flux B, which is calculated by finite element methods (FEM). We often use the FEMM tool in particular, but any FEM tool would suffice. The time and compute power to generate the mesh and to solve the magnetic field B on the mesh varies strongly with the resolution. And so does the precision. For our needed accuracy this library is much faster than before.
Still, FEM can generate problems, when the magnetic field is not exactly divergence-free. If the the authors of the magpylib library implemented the analytical expressions of the several papers mentioned on their website correctly, this is taken care of. The preciser solution comes from this library. We only tested if our calls and processing from library results are consistent with the analytical result for an especially parallel field. This is the case.
In the future, we will use this tool to calculate the field lines for the PIC simulation. Here, we only want to stress the point of faster and more accurate calculation of field lines in order to have a means for optimizing procedures like genetic algorithms, etc. These need to be fed with rather simple measures, which we try to derive on a geometric and physical basis.

[lines 200-201] "Comparing the different magnetic fields, one observes a protruding of the magnetic field into the vacuum outside the magnet.". Is this a comparison between Fig. 3 panels a, b, and c? If yes, please write this explicitly.

Answer:

Thanks for the suggestion. Indeed, we refer to these three geometries exemplarily. These three geometries are the extreme cases of the scan, where the length L is varied. The relative measure of the protruding magnetic field varies greatly between the short ring (a) and the long ring (c).
The text has been adapted accordingly.

[lines 201-202] "The z coordinate at the axial symmetry (r = 0) where the axial magnetic field Bz changes direction is a measure for the position zsep of the separatrix of the magnetic field.". I strongly suggest you showing z_sep in Fig. 3 panels (a, b, and c), for example, by adding on them a large point labeled z_sep.

Answer:

Good point. The marks are included. This will definitely help the readers.

[Figures 4 and 5] The font size in these figures is huge compared to the others. I suggest you adopting the same font sizes and styles for all figures.

Answer:

All plots are overhauled. The font size has been harmonized. Figures 4 and 5 have been combined in order to be more concise.

[lines 218-220] "The length L is varied between 10mm and 90mm in 2mm increments for a fixed R=20mm. Similarly, R is varied between 6mm and 40mm, also in steps of 2mm, while L is fixed at 50mm.For each of these cases, the field and the integration of the 20 field lines are performed, resulting in the subsequent figures.". Again, please address explicitly which figure depicts which case.

Answer:

The integrated field lines along the magnetic flux B are evaluated for each geometry, so that further analysis can be done. The placement was altered and the text rephrased. (lines 210-214)

"In the following, different geometries are calculated, where the length L is varied between 10mm and 90mm in 2mm increments for a fixed radius R=20mm.  For each geometry, the field and 20 selected field lines are computed.  Visually, this procedure is transcribed in Fig. 3. Two measures are introduced in order to characterize the geometry. The subsequent figures demonstrate different properties of the magnetic flux B."

[lines 234-241] Is Figure 6 originated from the same simulation that originated Figure 5 of Ref. 21? I understand that all details can be obtained from it ("Further details can be found in [21]"). However, I suggest you at least mentioning in this manuscript the PIC code you used for the simulation.

Answer:

Thank you for the tip. The code is developed within the group. A review of its applications can be found in the referenced review. The text is adapted. (lines 258-259)
"The code for the simulation was developed within the group.258Further details about the physics and the PIC code can be found in [23]."

[Figures 7 and 8] Besides having a huge font size, the legend keys overlapping half panel in these figures is a very poor choice. Please fix this.

Answer:

The font size of legend and axes is reduced as it is for the other plots. The legend keys have been moved.

[Conclusions] Please check the English/grammar writing in this section (some sentences and words are not correct).

Answer:

Grammar and language have been checked. Sentences should be improved.

Reviewer 2 Report

Dear Authors,

many thanks for providing this interesting paper on the optimization of ion thrusters. As a big space fan, I really hope your work will have an impact on future space exploration. Please find attached a few suggestions on how to improve the clarity of the exposition and the editorial quality of the manuscript.

Author Response

Dear Reviewer,

We are very grateful for your suggestions and really helpful comments. The recommendations helped us to improve the text, its readability and logical structure.

Answers to your questions can be found below. Most of the text changes are highlighted in blue. Please see the PDF for details.

Major technical issues
1) I do not find the whole concept of the optimization, which is the core of the paper, explained in a
sufficiently clear manner. There are many quantities of interest, namely: the length of the field
lines; their parallelism; the protrusion of the separatrix; the active volume ratio; the mirror ratio
and the confinement time. In the end, what is optimized ? One variable at a time ? A particular
combination ? The objective function of the optimization should be precisely defined. Or, if there
is none and this work is only an exercise to establish a method, then no problem, but it must be
stated clearly

Answer: This paper is merely an exercise to establish the method. We want to elucidate with parameter scans in R and L the effectiveness of simple performance indicators, which can guide the design of cusped field thrusters.
These indicators are presented. To get a global optimum for these quantities, one needs to verify the results with PIC simulations and finally experiments. This is out of the scope of this paper, but we will look into this.
We tried to clarify the objective in the text.
The field line length is an important parameter, because longer field line lengths are better for increased ionization probability for electrons. Together with the magnetic mirror effect at the cusps it allows to keep electrons longer within the volume and thus enhances ionization rates and efficiency. (lines 120ff)
The condition of a homogeneous field in the middle of one stage supports the design goal to limit the wall contact as much as possible to the cusps.

2) The degrees of freedom of the optimization seem to be only L and R. Why is the radial thickness
W of the magnet ring omitted ? This has obviously a big impact on the shape of the field lines.

Answer: Indeed, the thickness is also a very important parameter. We calculate a thin cylinder with a small width and focus on the variation of R and L, since we can relate this to the analytical estimate.
If the thickness increases, the field will become stronger, the magnetic ratio will increase so that the electron confinement time will rise. These are beneficial properties for the plasma. For a more extensive optimization one needs to put in also manufacturing and cost constraints, otherwise we expect, that the magnets will simply grow bigger. 
We plan to look at this in the future together with partners from industry, namely Thales.

3) Apart from a vague statement on line 48, no mention of the strength of the magnetic field is
made in the paper. The field strength obviously varies a lot, when L and R (and W) are changed.
Can you please elaborate on the impact of this quantity on your design ?

Answer:
The strength of the magnetization is set to a fixed value of µ0*M = 1000 mT. This is typical for Samarium Cobalt magnets. Only the front face enters in the Poisson equation as source with constant density. The area of the front face of the magnetic ring then scales only with the radius R. Thus, the total strength of the magnet scales with R, but also the distances for various field lines with starting point r_0 increase, so that the magnetic mirror ratio is not simply linearly growing.

4) Equation (3) is presented without derivation. Many similar or more general expressions can be
found in the literature, could you please quote references or give a little more detail about how
you derive it.

The potential of a ring charge is the same as in the electrostatic case Phi(rho) = k q/rho, where rho = sqrt(r^2+z^2) is the distance from the source and q is the total charge of the ring. k changes to mu0*M/(4*pi) and q becomes the magnetization M.

5) Line 140: based on (3), which is valid only at r=0, one can derive only the field component
Hz = -Φm/z, which contains contributions from all the coefficients appearing in (4), including
the 1st order one. Nothing can be said about the radial component Hr, which is the one that must
be reduced to guarantee parallel field lines. Therefore, I do not understand how zeroing the 3rd
order coefficient in (4) has anything to do with the stated goal. Could you please elaborate ?

The calculation can be found directly in Jungerman, where this is shown for the electrostatic case, which is analog to the magnetostatic case. 
The uniqueness theorem of the Poisson equation, states that the solution of the Poisson equation, in our case the magnetic potential, is determined up to a constant, so that the magnetic field H as the gradient of the potential is unique.
The potential in eq. 1 (the numbering has changed) is a function of z due to the symmetry.
The potential contains linear, cubic and fifth order terms in z. We aim to remove the cubic term by finding the root. Then, the solution depends on z and z^5.
A multipole expansion off-axis in r and rho, where rho = sqrt(r^2+z^2), in Legendre polynomials gives the r dependency.  In the linear term then r cancels and only a fifth order term in Legendre polynomials remains. Due to the expansion, the potential has terms depending on r and z in various orders. The removal of the linear and cubic terms thus minimizes the field in r.
We modified the text accordingly and can provide a derivation if necessary. The calculation could be put into the appendix. 

6) Line 280: the quoted optimal ratio L/R = 5/1 is not consistent with the figure 2.45 quoted on line 153.

Thank you very much for pointing this out. We clarified this statement. The analytical result only gives a ratio of length to radius for one stage. In the conclusion we are referring to a typical two-stage thruster, which then doubles this number.

Other technical issues
7) line 60: “the cost of the permanent magnets can be high”. Could you please clarify what you
mean ? I would expect the material cost of PM to be a small fraction of the cost of
manufacturing and operating an equivalent electromagnet. (especially because later you work
on the assumption of infinitesimally thin magnet rings ...)

Answer:
The infinitesimal thin magnet is needed for the analytical derivation. Real magnets with finite width can add a significant share to the costs. They are made of rare earth metals, which increased in price in recent years. The further application of this model will use then parameters of the real magnets used in industry without the approximation.

8) The nature of the metal parts surrounding the magnets should be specified, in particular is there
any that is ferromagnetic, which would affect the field lines ?

Answer:
The metal parts are usually made of stainless steel, which can be ferromagnetic or not, but can be included in the model. Unfortunately, some of the information is blocked by a NDA.
In Fig. 6 the area marked with the brown square is a grounded metal. The plasma is also influenced by radial electric fields created by this boundary condition, but also from internal fields created by the plasma.

9) Line 140: the potential in Eq. (3) is not “at the center O of stage” but rather “on the axis of the
stage (
=0)”

Answer:
We corrected the text according to your suggestion.

10) Line 142/143: the referenced variables a, d and c appear nowhere in (3).
Also, the sentence “the front faces are reduced to a point with distances in a axial direction z” is
unintelligible.

Thank you for pointing this out. We had some typos.

11) The geometry of the stage is described in the text by two sets of variables, (L,R) and (a=R,
d=L/2). This is unnecessary and confusing. I would suggest sticking to L and R, which are self-
explanatory and clearly defined in Fig. 2

Answer:
The corresponding text is corrected.

12) In the text and figures ρ seems to be used interchangeably with r. Please stick to a unique
definition.

Answer:
Again, we corrected the notation. We hope that the new notation is consistent.

13) Fig. 2: the orientation of the magnetization vector is not consistent with the field lines. Both B
and H go from magnetic North to South poles. Please indicate which pole is which, for
consistency with Fig. 1.
In addition: I think that the radial thickness w of the magnet ring should be indicated here.

Answer:
We changed Fig. 1 to be consistent with the other figures. The right pole is now the North pole. Arrows of the field lines are also updated.
We removed the thickness w from the code and just name the finite magnet width of 2mm. We do not scan the width here, which would also be an interesting parameter to analyze, but is out of the scope for this paper.

14) Fig. 3: “Shown is only a quarter of the magnetic system, since the magnet is also symmetric with
respect to the axial symmetry axis (r = 0)”. This statement is incorrect. The figure shows only one
half of the axially symmetric system, the missing half being at negative z. This profits from
symmetry respect to the plane (z=0). Axial (or cylindrical, they are synonyms) symmetry takes
care of the 3D cylindrical volume, and it does not make sense to speak of another “”bottom
half”.

Answer:
The text is adapted as you suggested.

15) Line 201-208: the definition of the quantity plotted in Fig. 4 is clear, but why is it important ? Is it
an optimization objective or not ? Do you want it to be large or small ? Please elaborate.
For more clarity, zsep should also be shown in Fig. 2, 3 and 6

With the protrusion of field lines the magnetically bound electrons might follow. Due to quasi-neutrality condition in a plasma the ions are coupled to the electrons. The plasma is then pushed outwards. Eventually, the plasma shielding will break down if the densities drop because of the wall losses. If this happens too far outside the thruster channel, the electric field will have undesirable radial components, which will worsen the thruster performance. To what extent higher angles will be generated is out of the scope of this paper. This needs proper checking and testing with kinetic simulations like PIC. 
More details about the importance of the electric field structure at the exit of the thruster for the ion angular distribution can be found in Duras et al., Ion angular distribution simulation of the Highly Efficient Multistage Plasma Thruster.Journal of Plasma Physics2017,83, 595830107.  https://doi.org/10.1397017/s0022377817000125.
or in
Matthias et al., Particle-in-cell simulation of an optimized high-efficiency multistage362plasma thruster.Contributions to Plasma Physics2019,59, e20190002

We have added a marker in Fig. 3. We do not plot a marker in the sketch of Fig. 2, since the field lines are not representative.
In Fig. 6 one can see a sharp drop in the electron density at the end, which is equivalent to radial fields. Plotting here z_sep is possible and might be helpful.

16) Line 228: “The integration of several field lines at equidistant radii ρ delivers a length scale
measure of the magnetic field”: this sentence is not clear, how is exactly this length scale
defined from multiple field lines ? Is it just the average ?

Yes, the blue lines are just the average in Fig. 7 and 8. Also 3 selected field lines with relative starting position r0/R are plotted to see the variation of field line lengths for the different geometries. Generally, the average seems to be a good indicator for \ell, R_m and tau. 
The magnetic mirror R_m adds reflection to the cusps, so that the path of electrons can increase. For the confinement time additional radial and density scaling is applied. The longer the electrons stay within the channel the more they can ionize neutrals and increase the plasma density.

17) Line 229: “For this If we increase the length L, the magnets become longer and thus the
magnetic field lines”: this obvious fact does not appear to be logically related to the previous
sentence. If you just want longer field lines, then why don’t you simply use very long magnets ?

Answer:
The text is adapted.  It reads now:
"The field line length is nearly proportional to the length L. For small magnet lengths, the lengths $\ell$ deviate from a linear dependence, since the bigger elongation in the z direction adds to the path."

18) Fig. 6: could you please show the width of the magnet in the figure ? Otherwise, one might think
it is the whole gray rectangle, which would contradict the approximation of thin magnet used to
derive equation (3). If that is the case, please elaborate.

Answer:
The gray area is not the magnetic geometry, it just indicates that this area is not accessible to particles in the PIC simulation, because it is blocked in radial direction by a dielectric and a metal front blade (indicated in brown).
The caption of Fig. 6 now reads:
"Electron density n_e from a 2D PIC simulation of a two-stage DM3a HEMP thruster in the (r,z)-plane. The rectangular regions in gray and brown indicate a dielectric and metal wall, respectively. Particles are prohibited in these areas."

19) Line 244: “Because of the cylinder symmetry, field lines starting at the radial coordinate r0 scale
with the circumference and are r0 times more frequent.”: this sentence is not at all clear to me.
A field line is not a number and cannot scale with anything. You must be referring to some
quantity attached to field lines, which one ? And what do you mean by frequency ? Is it a
frequency measured in Hz ? I suspect you are looking for a different English word.

Answer:
If we can increase the length at greater radii r0. These field lines occur more often.
We changed the text to:
"Because of the cylinder symmetry, number of field lines starting at the radial coordinate r0 scales with the circumference and are r0 times more abundant."
This not to be confused with the confinement time tau  of the electrons as defined in the paper.

20) Line 255: the symbol  for the length of magnetic field lines, which is an important quantity
referenced throughout the paper, is introduced only here at the end, which sounds a bit odd.
Could you please define it and use it at an earlier stage ? A precise definition is needed including
integration bounds. Do you use H or B ? Do you stop at the ideal wall r=R, or follow the whole
closed B line, or follow H from pole to pole ?

Answer:
The definition for \ell is put further towards the beginning of the paper.

21) Fig. 7: parametrization of the curves: this is not clearly defined, is it the radial starting point r0 ?
If so please state it clearly. If the legend is the same for all figures it would be best plotted
separately, or repated. And, rather than use millimeters, I think that using a fraction of R would
be more meaningful.

Answer:
The figure was changed and relative lengths are used. 

22) Fig. 7: why do you use arbitrary units for the magnetic length ? Would it not be more meaning
meaningful, for example, to use units of L

Answer:
You are absolutely right. The length is in mm. We will adjust the label.

Minor language and formatting issues
Line 53: the units in “1
N” and “100mN” are italicized and attached to the preceding number. They
must be written in straight font and separated by a non-breaking space. These errors are repeated many
times throughout the text, please do review it and correct carefully.
Line 113/114: “The code is written in the Python programming language ...” this should be stated before
the reference to Python made on line 109
line 140: “0” -> should be an O”
line 151: variables such as d, a and c should be in italics, here and everywhere else throughout the paper
lien 155: acronym PIC should be defined here, not on line 170. Also use it on line 232 for consistency
line 181: why rho is not written as the Greek character
 ?
Answer:
rho should not be used here at all. 

line 181/185: “simplified”→ “ideal”
line 181: “imaginative”→ “ideal”. “imaginative” is the quality of a human being with a lot of fantasy.
line 186: do not break the equation over two lines
line 201: “at the axial symmetry” → “on the axis”. Your system has only one axis, unambiguously defined
at the beginning of the paper.
line 202: “is a measure for” → “defines”
line 229: “If” → “if” lowercase
line 230: “thus” → “so do”
line 256: “radial axis” → “radial direction”. There only one axis in a cylindrically symmetric reference
system, and that is the z axis (r=0).
Fig. 3 caption: R = 20 mm please add spaces between numbers and units. Also applies at many other
places in the document

Answer:
Thank you very much for the detailed response. We will adapt the text according to your above suggestions for the minor language and formatting issues. Your comments are invaluable!

Fig. 3 and all the following ones: in “/ mm” the units should be written between round brackets “(mm)”.
The slash character can be confused with the division symbol.

Answer: In plots and tables numerical values are presented so that the unit is divided from the physical quantity. NIST is suggesting this notation. Please see the details in section 7.1 on https://www.nist.gov/pml/special-publication-811/nist-guide-si-chapter-7-rules-and-style-conventions-expressing-values
In case you or the editors want a different style we will adapt the figures.

Reviewer 3 Report

1. The abstract is written too simply and needs to be supplemented with the main research steps and key conclusions of the optimization work.

2. The author mentions that optimizing the performance of the thruster can be challenging due to the complexity. However, when the optimization of the structure is done in the text, the optimized structural parameters are very simple, just R and L parameters. In the introduction, it is mentioned several times that the optimization objectives should take into account the design of the ionization chamber, the acceleration phase and the overall operating conditions. Specifically, for example: 1) the ionization chamber is designed to maximize the ionization rate while minimizing the energy required. 2) the acceleration of the plasma is maximized, and the energy required to accelerate the plasma is minimized. 3) the chamber should be designed to minimize the number of nonconforming neutral particles as much as possible. However, the above factors are also not fully considered in the study of this paper. 3. the paper lacks a description of the specific optimization methods used.
4. Figure 6 is definitely not a theoretical analytical solution, what software was used to calculate it? It needs to be described.

5. The format of the references is not quite standard.

6. The thesis work is relatively simple and lacks novelty.

Author Response

Dear Reviewer,

Thank you for your suggestions. We tried to improve the paper according to your constructive criticism.

Answers to your questions can be found in the text below. Most of the text changes are highlighted in blue. Please see the PDF for details.

1. The abstract is written too simply and needs to be supplemented with the main research steps and key conclusions of the optimization work.

Answer:
The abstract is modified. The findings are described in more detail.

2. The author mentions that optimizing the performance of the thruster can be challenging due to the complexity. However, when the optimization of the structure is done in the text, the optimized structural parameters are very simple, just R and L parameters. In the introduction, it is mentioned several times that the optimization objectives should take into account the design of the ionization chamber, the acceleration phase and the overall operating conditions. Specifically, for example: 1) the ionization chamber is designed to maximize the ionization rate while minimizing the energy required. 2) the acceleration of the plasma is maximized, and the energy required to accelerate the plasma is minimized. 3) the chamber should be designed to minimize the number of nonconforming neutral particles as much as possible. However, the above factors are also not fully considered in the study of this paper. 

Answer:
Indeed, the paper introduces a rather simple method for calculating the magnetic field configurations of HEMP thrusters. This is the key idea of the publication: replacing a complicated numerical solution procedure (usually Finite-Element codes like FEMM) by a fast, analytically based method, which has in addition the advantage that due to the analytical solution divergence-free resulting magnetic fields are available for e.g. Particle-in-Cell models. FEMM needs rather high resolutions, resulting in long calculation times and rather big input data sets, to avoid artificial drifts from numerical artefacts. This method has the nice built-in feature that this is guaranteed by the analytic approach. As an example for the possibility of using this analytical approach for the optimization work for HEMP thrusters, the optimized ratio of radius to length is calculated analytically using an expansion. The result agrees very well with published experimental and numerically optimized designs (using genetic algorithms). This allows a better understanding of these optimization results, which were not explained before. 
This example is then used to validate the existing python package and proof its validity for further applications. Using for the analytically calculated magnetic fields additional diagnostics motivated by physics considerations demonstrates then the basic idea how to implement this tool into optimization procedures for HEMP thrusters.
Here, we can already highlight that the length should be longer than the prediction from the analytical expansion.

3. the paper lacks a description of the specific optimization methods used.

Answer:
We use an analytical result for a parallel magnetic field. We calculate the length of certain, selected field lines in order to calculate performance indicators, which describe the length and form of field lines. We calculate different geometries, compute the measures. Alone from the figures 5, 7 and 8, which show these quantities, one can find a better magnetic geometry than the simple analytical result. 
It is clear that this can not be the end. The problem is quite complex due to the interplay of the charged species, the magnetic field and the boundaries. Here, we want to address the magnetic field, since we have an analytical estimate and a helpful library. The optimization of a whole thruster is much more work. In the end, a result of the optimization needs to be checked with a fully kinetic simulation such as PIC and finally by experiments. In our opinion, the optimization becomes feasible with fast and simple measures of the magnetic field, which are here presented.

4. Figure 6 is definitely not a theoretical analytical solution, what software was used to calculate it? It needs to be described.

Answer:
The figure contains the results from a previous Particle-in-Cell (PIC) simulation. The PIC code was and still is developed within the group. The text is amended. A reference to a review paper of the code and its application is given.

5. The format of the references is not quite standard.

Answer:
Most of bibtex entries are generated from the doi or are directly downloaded from the journal.
We fixed the bibtex references for the HEMP patents and repaired some minor problems.
Here, we just follow the standard style of the journal. We use the provided Latex template of MDPI Applied Sciences. No alterations to the style template were performed.

6. The thesis work is relatively simple and lacks novelty.

Answer:

In this work we show analytically for a certain ratio of R and L that the field is particularly parallel at the origin. This property itself is a good feature for these types of thrusters. Ongoing from this result, we use an existing open-source library, which can calculate analytically the magnetic field for specified positions. As a first result we show that the analytical and numerical fields coincide, when comparing the ratio of radial to the total magnetic field B in Fig. 4. This is to assert the correctness of either our expansion or the results of the library.
Then, for a range of different geometries certain measures are introduced, which we believe describe features of the magnetic field, which are crucial for thruster performance. The idea of this paper is to provide simple measures which can be employed in an optimization process. Here, we can already highlight that the length should be longer than the prediction from the analytical expansion. We are aware that these measures are oversimplified. The idea is to find a configuration space where it is worth to check the performance of the thruster by a computationally costly PIC simulation. This library facilitates the computation by its speed and also by its accuracy. Because of this we will use this as a tool in the future to compute the magnetic field instead of finite element methods (FEM). FEM tools can generate problems, when the magnetic field is not exactly divergence-free.
The simple expressions should be a basis for the optimization of the thruster. The magnetic field is only one aspect. One can for instance change the material and geometry of the channel wall, applied voltages, etc. Here, we only tried to tackle one problem at a time. Of course, this is not the global optimum.
Interestingly, for a thruster consisting of two stages a ratio of length to radius of 5 to 1 is prevalent in industrially designed thrusters and calculations from previous optimization processes. This reassures the results we present here.

Round 2

Reviewer 3 Report

The paper lacks sufficient depth in terms of novelty and theory, but the authors respond to all questions and answer them in a generally reasonable manner.